# Nerve growth factor receptor negates the tumor suppressor p53 as a feedback regulator

Xiang Zhou[1,2†‡], Qian Hao[1,2†§], Peng Liao[1,2], Shiwen Luo[3], Minhong Zhang[3], Guohui Hu[3], Hongbing Liu[1,2], Yiwei Zhang[1,2], Bo Cao[1,2], Melody Baddoo[2], Erik K Flemington[2], Shelya X Zeng[1,2], Hua Lu[1,2*]

[1]Department of Biochemistry and Molecular Biology, Tulane University School of Medicine, New Orleans, United States; [2]Tulane Cancer Center, Tulane University School of Medicine, New Orleans, United States; [3]Center for Experimental Medicine, The First Affiliated Hospital of Nanchang University, Nanchang, China

**Abstract** Cancer develops and progresses often by inactivating p53. Here, we unveil nerve growth factor receptor (NGFR, p75NTR or CD271) as a novel p53 inactivator. p53 activates NGFR transcription, whereas NGFR inactivates p53 by promoting its MDM2-mediated ubiquitin-dependent proteolysis and by directly binding to its central DNA binding domain and preventing its DNA-binding activity. Inversely, NGFR ablation activates p53, consequently inducing apoptosis, attenuating survival, and reducing clonogenic capability of cancer cells, as well as sensitizing human cancer cells to chemotherapeutic agents that induce p53 and suppressing mouse xenograft tumor growth. NGFR is highly expressed in human glioblastomas, and its gene is often amplified in breast cancers with wild type p53. Altogether, our results demonstrate that cancers hijack NGFR as an oncogenic inhibitor of p53.

*For correspondence: hlu2@tulane.edu

[†]These authors contributed equally to this work

Present address: [‡]Shanghai Cancer Center and Institutes of Biomedical Sciences, Fudan University, Shanghai, China; [§]Shanghai Cancer Center, Fudan University, Shanghai, China

Competing interests: The authors declare that no competing interests exist.

## Introduction

Tumorigenesis is highly associated with inactivation of the tumor suppressor p53, as it is mutated in ~50% of all types of human cancers, and its functions are impaired through various mechanisms in the rest of human cancers (*Levine and Oren, 2009*). p53 executes its tumor suppressive function mainly by inducing the expression of a large number of genes involved in cell cycle control, senescence, apoptosis, ferroptosis, autophagy, and metabolism (*Jiang et al., 2015*; *Kruiswijk et al., 2015*). Because of the detrimental effects of p53 on normal cells, the cells have developed mechanisms to monitor its activity, which are often hijacked by cancer cells.

One key monitor of p53 is MDM2 (HDM2 in human), an oncoprotein encoded by a p53 transcriptional target gene that is amplified or overexpressed in several human tumors (*Fakharzadeh et al., 1991*; *Momand et al., 1992*; *Oliner et al., 1992, 1993*; *Wu et al., 1993*). MDM2 inhibits p53 activity primarily by binding to and concealing the N-terminal transcriptional activation (TA) domain of p53 (*Oliner et al., 1993*) and by mediating its poly-ubiquitination and proteolysis (*Fuchs et al., 1998*; *Haupt et al., 1997*; *Kubbutat et al., 1997*). Genetically, disruption of the *TP53* gene completely rescues the lethal phenotype of Mdm2 knockout mice (*Jones et al., 1995*; *Montes de Oca Luna et al., 1995*). A myriad of stresses can orchestrate this MDM2-p53 feedback loop. The ARF tumor suppressor directly associates with MDM2 and inhibits MDM2-mediated p53 ubiquitination and degradation upon oncogenic stress (*Palmero et al., 1998*; *Zhang et al., 1998*; *Zindy et al., 1998*). Also, several ribosomal proteins boost p53 activation by untying the MDM2-p53 loop in response to ribosomal or nucleolar stress (*Zhang and Lu, 2009*; *Zhou et al., 2012, 2015a*). But,

**eLife digest** Cancer often develops as a result of alterations to "tumor suppressor" genes within cells. This results in the cells growing and dividing too much, which causes a tumor to form. One of the most important tumor suppressor genes produces a protein called p53, which is lost or mutated in roughly half of all human cancers. In the other half of cancers p53 itself is normal, but is often disabled by proteins that promote tumor growth.

One of the remaining challenges in the field of cancer research is to identify which proteins inhibit p53 directly. Identifying these proteins would help clarify why many human cancers, such as some brain cancers, breast and skin cancers, often maintain a normal form of the p53 tumor suppressor protein.

Zhou et al. now provide evidence that shows that a protein called nerve growth factor receptor (NGFR) is one such protein. NGFR was known to be important for the healthy development of the brain and nervous system. Unexpectedly, however, Zhou et al. found that NGFR binds directly to p53 and disables it in several types of human cancer cells. This finding is likely to be important because NGFR is produced in abnormally high amounts in several human cancer types, including skin, breast, bone and some brain cancers.

Reducing the levels of NGFR in cancer cells caused the cells to become more sensitive to some anti-cancer drugs. Overall, the results presented by Zhou et al. suggest that developing new drugs that target NGFR could produce new treatments for human cancers that have a normal form of the gene that produces p53. More experiments are also needed to find out whether NGFR has other ways of promoting the development of cancerous tumors.

oncogenic proteins can enhance MDM2 E3 ligase activity towards p53. MDMX (also called MDM4), the MDM2 homologue, can enhance MDM2-mediated p53 proteasomal degradation by binding to MDM2, besides directly interacting with p53 and repressing its activity (*Shvarts et al., 1996*). High expression of MDM2 and MDMX in several cancers, such as breast cancer and melanoma, is often considered as the reason why these cancers sustain wild type (wt) p53 (*Wade et al., 2013*), but this could only account for a portion of wt p53-harboring cancers. Thus, it is still unknown if there are other proteins that can also suppress p53 function in the remaining cancers.

In this study, we revealed a novel feedback regulation of p53 by nerve growth factor receptor (NGFR, also called p75NTR or CD271). NGFR is a 75 kD single-transmembrane protein without kinase activity and widely expressed in the central and peripheral nervous system (*Barker, 2004*). Often partnering with other receptors, such as TrkA, it is involved in a multitude of processes during neurogenesis, such as neural cell death, neuronal differentiation, neurite growth, and synaptic plasticity (*Barker, 2004*). Also, the NGF-NGFR cascade activates NF-κB, leading to inhibition of apoptosis (*Carter et al., 1996*) and increased survival of schwannoma (*Ahmad et al., 2014*; *Gentry et al., 2000*) and breast cancer cells (*Descamps et al., 2001*). In addition, overexpression of NGFR observed in many metastatic cancers promotes tumor migration and invasion (*Boiko et al., 2010*; *Civenni et al., 2011*; *Johnston et al., 2007*). But, in prostate and bladder cancers, NGFR appears to suppress tumor growth and/or metastasis (*Krygier and Djakiew, 2002*; *Tabassum et al., 2003*). It remains largely elusive why and how NGFR plays opposite roles in the context of different cancers.

These studies together with our initial findings that p53 binds to the *NGFR* promoter and induces its expression in cancer cells motivated us to further explore the functional interplay between NGFR and p53, and its role in cancer development. As detailed below, we surprisingly found that NGFR inactivates p53 by directly binding to its central DNA-binding domain and preventing its association with its target promoters and by enhancing its MDM2-mediated ubiquitination and proteolysis. This function is ligand-independent because it occurred in the nucleus and without ligand treatment of cancer cells. Biologically, cancer cells hijack the negative feedback regulation of p53 by NGFR to their growth advantage, as down regulation of NGFR induced p53-dependent apoptosis and cell growth arrest as well as suppressed tumor growth. Furthermore, NGFR was found to be highly expressed in 68.75% (33/48) of human gliomas examined. Consistently, NGFR is amplified in breast cancers that harbor wt TP53 based on the TCGA database (*Cerami et al., 2012*; *Gao et al., 2013*).

Hence, our discovery of NGFR as another feedback suppressor of p53 could explain why some cancers sustain wt p53 and also suggest NGFR as a potential target for the development of new anti-cancer therapy.

## Results

### *NGFR* is a bona fide transcriptional target of p53

From our previous studies to assess the global effects of Inauhzin (INZ) on p53 pathway in cancer cells (*Zhang et al., 2012*, *2014*; *Liao et al., 2012*), we identified *NGFR* as a potential p53-regulated gene. To confirm this result, we treated three types of p53-containing cancer cell lines (HCT116$^{p53+/+}$, H460 and HepG2) with INZ, Doxorubicin (Dox) and 5-Fluorouracil (5-FU). The expression of *NGFR* mRNA was drastically elevated by all the three agents (*Figure 1A,B and C*). Consistently, NGFR protein level increased in response to Dox or 5-FU treatment in p53-intact, but not p53-null (HCT116$^{p53-/-}$) or mutated (PCL/PRF/5), cancer cells (*Figure 1D and E*). Consistently, ectopic wt, but not mutant, p53 induced NGFR mRNA expression in p53-deficient H1299 and HCT116$^{p53-/-}$ cells (*Figure 1F and G*). Conversely, knockdown of p53 markedly reduced *NGFR* mRNA level (*Figure 1H and I*). These results demonstrate that anti-cancer drug-induced NGFR expression in the cells is p53-dependent.

Next, we identified two potential p53 responsive DNA elements (RE) in the *NGFR* promoter by bioinformatics analysis, RE1 and RE2, at −4826 bp and −6516 bp upstream from the transcriptional initiation site, respectively (*Figure 1J*). The RE1 was responsive to p53, as ectopic p53 markedly activated luciferase reporter expression through RE1, but not RE2, in H1299 (*Figure 1K*) and U2OS (*Figure 1L*) cells. This result was validated by chromatin-associated immunoprecipitation (ChIP) assays after treating HCT116 $^{p53+/+}$ cells with or without Dox. Endogenous p53 specifically associated with the RE1-containing *NGFR* promoter, but not a non-related promoter; this association was enhanced by Dox treatment (*Figure 1M and N*). Thus, these results demonstrate that *NGFR* is a bona fide transcriptional target of p53.

### NGFR is required for cancer cell survival and clonogenicity

Previous studies showed that NGFR induces apoptosis in some cancer cells, but promotes cell survival and growth in other cancer cells (*Molloy et al., 2011*). Thus, we re-determined the role of NGFR in cancer cell growth and proliferation and whether this role is p53-dependent. Surprisingly, knockdown of NGFR elicited significant apoptosis of H460 (*Figure 2A and B*) and HCT116 $^{p53+/+}$ (*Figure 2C and D*) cells as measured by flow cytometric analyses, consequently reducing cell viability during the 4-day culture (*Figure 2E and F*). To verify if NGFR supports tumor cell growth and proliferation and to see if this role is p53-dependent, we performed colony formation assays by knocking down NGFR in p53-containing H460 and p53-deficient H1299 cancer cells (*Figure 2—figure supplement 1A*). NGFR ablation more dramatically repressed the colony formation of H460 cells than that of H1299 cells (*Figure 2G*). This result suggests that NGFR may support cancer cell survival through both p53-dependent and independent mechanisms. Since NGFR was found to be required for nerve growth factor-mediated NF-κB activation (*Carter et al., 1996*), knockdown of NGFR in H1299 may impair NF-κB activation, thus leading to partially restrained cell viability and reduced colonies. Additionally, immunohistochemical (IHC) staining of human gliomas and their adjacent normal tissues exhibited hyperexpression of NGFR in the tumor tissues (*Figure 2H* and *Figure 2—figure supplement 1B*). Consistently, NGFR was more highly expressed in human glioma tissues compared to adjacent tissues as determined by immunoblotting (IB) (*Figure 2I*). Taken together, these results suggest that NGFR promotes cancer cell growth and proliferation, likely in part by inactivating p53.

### NGFR inactivates p53 by enhancing its MDM2-mediated ubiquitination and proteolysis

To test if NGFR negates p53 activity, we first checked if NGFR affects p53 protein level in H460 cells that were treated with Dox or 5-FU. Indeed, p53 level induced by Dox or 5-FU was markedly reduced by ectopic NGFR (*Figure 3A*). Inversely, knockdown of NGFR strikingly elevated the level of endogenous p53 and that of its target genes p21 and PUMA in human p53-containing H460, HepG2, neuroblastoma SK-N-SH, melanoma SK-MEL-103 and SK-MEL-147 and HCT116 $^{p53+/+}$ cell

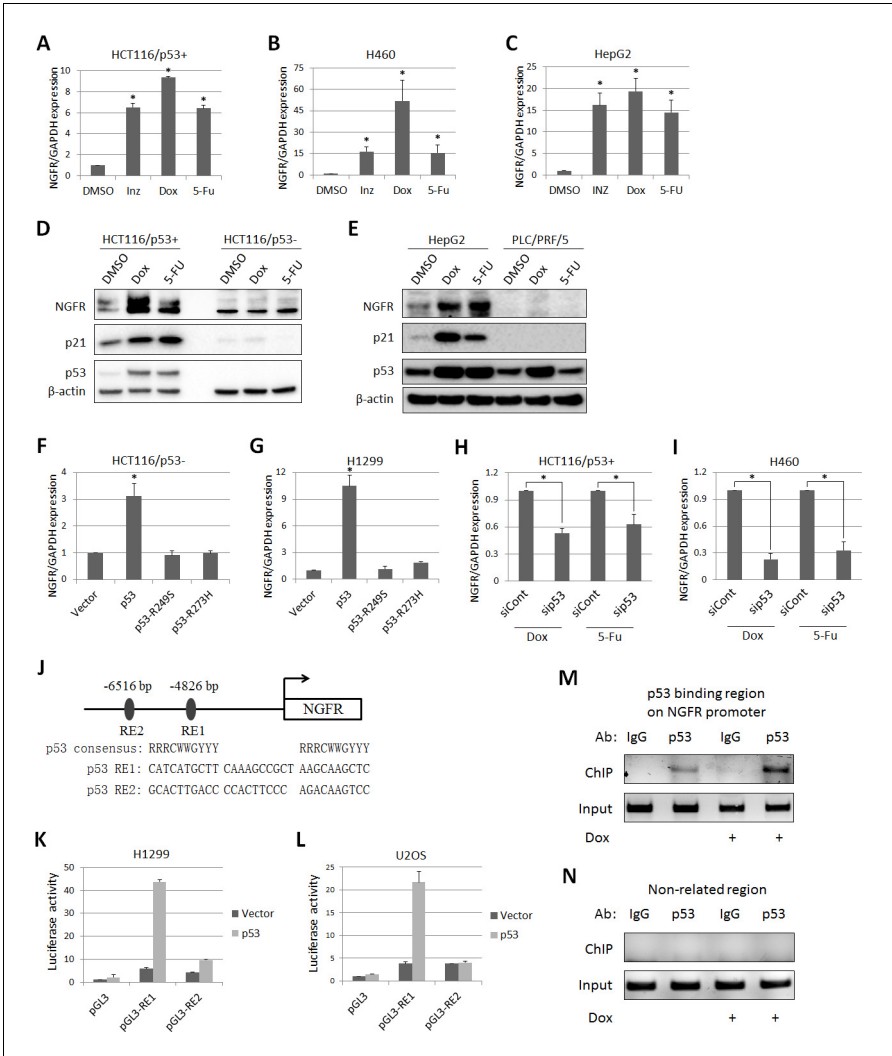

**Figure 1.** p53 transcriptionally induces NGFR expression in cancer cells. (**A,B,C**) NGFR mRNA expression is elevated by p53-inducing agents. HCT116 $^{p53+/+}$ (**A**) H460 (**B**) and HepG2 (**C**) cells were treated with Inauhzin, Doxorubicin or 5-Fluorouracil for 15 h, and NGFR expression was determined by q-PCR (Mean ± SEM, n = 3). Three biological replicates (independent experiments) and a two-tailed t-test were used for P value calculation, p*<0.01. Most q-PCR were performed by three biological replicates, each including three technical replicates. (**D**) NGFR protein expression is elevated by p53-inducing agents in colon cancer cell lines. HCT116 $^{p53+/+}$ and HCT116 $^{p53-/-}$ cells were treated with Doxorubicin or 5-Fluorouracil for 15 hr followed by IB using antibodies as indicated. (**E**) NGFR protein expression is elevated by p53-inducing agents in liver cancer cell lines. HepG2 and PLC/PRF/5 cells were treated with Doxorubicin or 5-Fluorouracil for 15 hr followed by IB using antibodies as indicated. (**F,G**) NGFR mRNA expression is induced by ectopic wild-type, but not mutant, p53. HCT116 $^{p53-/-}$ (**F**) and H1299 (**G**) cells were transfected with wild-type or mutant p53 for 30 hr and NGFR expression was determined by q-PCR (Mean ± SEM, n = 3). Three biological replicates were used for p value, p*<0.01. (**H,I**) NGFR mRNA expression is inhibited by p53 knockdown. HCT116 $^{p53+/+}$ (**H**) and H460 (**I**) cells were transfected with p53 or control siRNA for 72 hr, and Doxorubicin or 5-Fluorouracil was supplemented 15 hr before the cells were harvested for q-PCR (Mean ± SEM, n = 3). Three biological replicates were used for p value, p*<0.05. (**J**) A schematic of predicted p53 responsive elements in the NGFR promoter. (**K,L**) p53 induces luciferase activity through RE1. Luciferase assay was performed using H1299 and U2OS cells as described in Materials and methods (Mean ± SEM, n = 3 biological replicates). (**M,N**) p53 is associated with the NGFR promoter. HCT116 $^{p53+/+}$ cells were treated with or without Doxorubicin for 15 hr followed by ChIP assay using anti-p53 or mouse IgG.

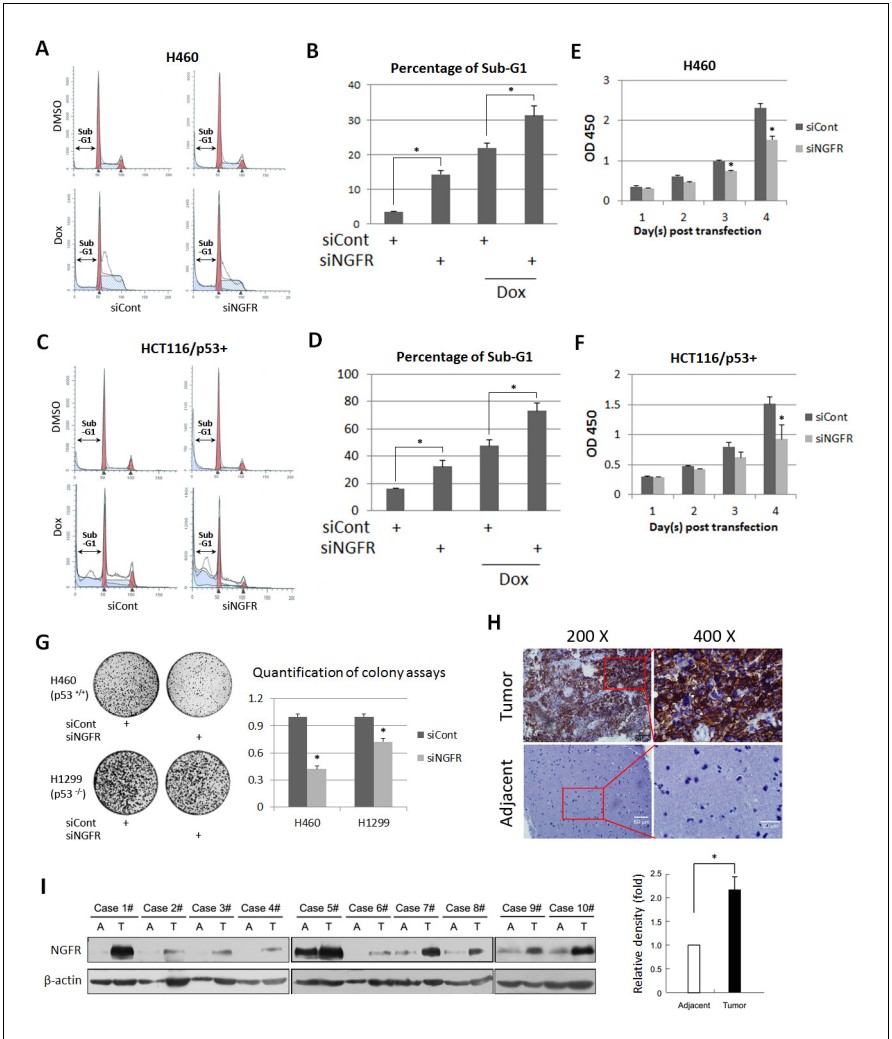

**Figure 2.** NGFR is required for cancer cell survival and clonogenicity, and highly expressed in glioma. (**A**) NGFR knockdown induces apoptosis of H460 cells. Cells were transfected with NGFR or control siRNA for 72 to 96 hr, and Doxorubicin was supplemented 15 hr before the cells were harvested for flow cytometry analyses. (**B**) Quantification of Sub-G1 population in (**A**) (Mean ± SEM, n = 3). Three biological replicates were used for p value, p*<0.05. (**C**) NGFR knockdown induces apoptosis of HCT116 $^{p53+/+}$ cells. Cells were transfected with NGFR or control siRNA for 72 to 96 hr, and Doxorubicin was supplemented 15 hr before the cells were harvested for flow cytometry analyses. (**D**) Quantification of Sub-G1 population in (**C**) (Mean ± SEM, n = 2). Three biological replicates were used for p value, p*<0.05. (**E, F**) NGFR knockdown suppresses cell survival. H460 (**E**) and HCT116 $^{p53+/+}$ (**F**) cells were transfected with NGFR or control siRNA and seeded in 96-well plate the next day (Day 1). Cell viability was evaluated every 24 hr (Mean ± SEM, n = 3). Three biological replicates were used for p value, p*<0.05. (**G**) NGFR knockdown inhibits clonogenicity. H460 and H1299 cells were transfected with NGFR or control siRNA and seeded in 10-cm plates the next day. Colonies were fixed by methonal and stained with crystal violet solution (left panel). Quantification of colonies is shown in the right panel (Mean ± SEM, n = 2). Two biological replicates were used for p value, p*<0.05. (**H**) IHC analyses of human glioma and the adjacent noncancerous tissues. (**I**) IB analyses of human glioma and the adjacent noncancerous tissues (left panel). Quantification of NGFR expression (right panel) (Mean ± SEM, n = 48, p*<0.01).

The following figure supplement is available for figure 2:

**Figure supplement 1.** Representative expression of NGFR in lung cancer cell lines by siRNA knockdown and in glioma tissues.

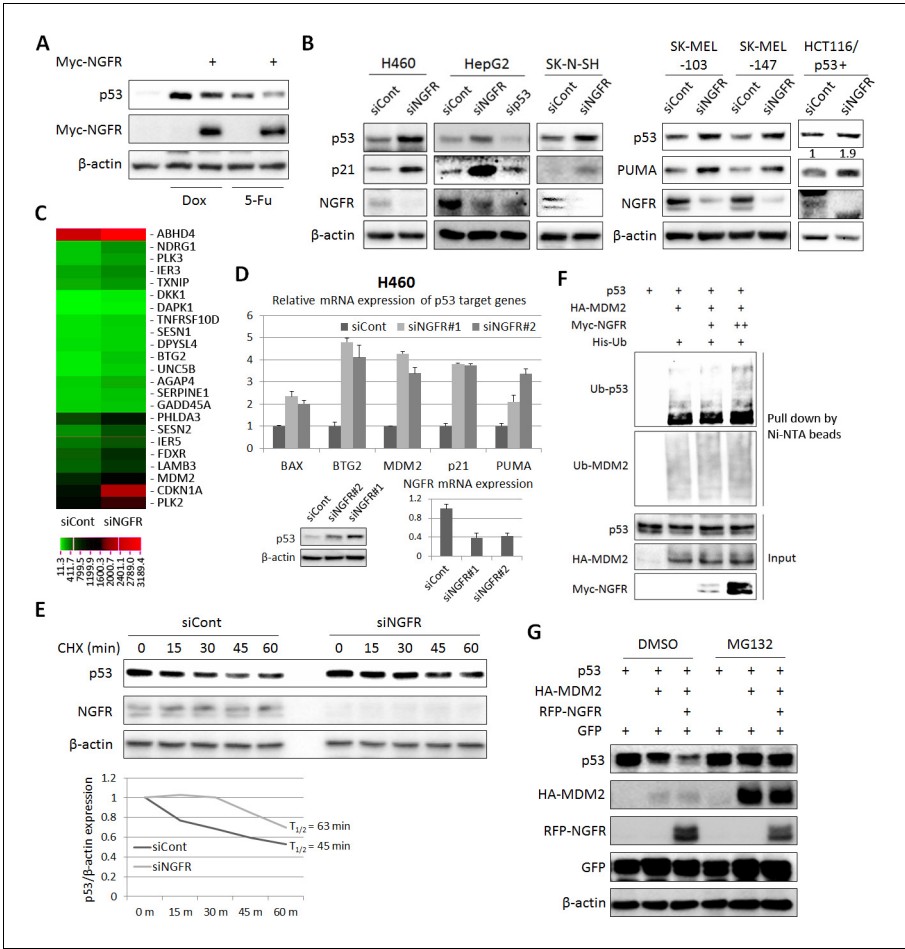

**Figure 3.** NGFR suppresses p53 activity by enhancing MDM2-mediated ubiquitination and proteasomal degradation. (**A**) NGFR inhibits p53 activation by Doxorubicin or 5-Fluorouracil. H460 cells were transfected with NGFR for 30 hr and treated with Doxorubicin or 5-Fluorouracil for 15 hr before harvested for IB using antibodies as indicated. (**B**) NGFR knockdown induces p53 expression and activity. A panel of cancer cell lines were transfected with NGFR or control siRNA followed by IB using antibodies as indicated. The values in the rightmost panel indicate the p53/β-actin ratios. (**C**) NGFR knockdown induces p53 target gene expression. H460 cells were transfected with NGFR or control siRNA followed by RNA-sequencing analyses. Genes with over 1.5-fold increase in expression were shown (Three biological replicates were used for p value, p<0.05. (**D**) Two siRNAs targeting different sequences were used for knocking down NGFR in H460 cells. The expression of p53 target genes was assessed by q-PCR (Mean ± SEM, n = 2 biological replicates), while p53 expression was detected by IB. Validation of NGFR knockdown by the two siRNAs is shown in the right corner. (**E**) NGFR knockdown prolongs p53's half-life. H460 cells transfected with NGFR or control siRNA for 72 hr were treated with 100 µg/ml of CHX and harvested at the time points as indicated. IB was performed using antibodies as indicated (upper panel) and quantification of p53/β-actin ratio is shown in the lower panel. (**F**) NGFR promotes MDM2-induced p53 ubiquitination. H1299 cells were transfected with combinations of plasmids encoding p53, HA-MDM2, Myc-NGFR or His-Ub and treated with MG132 6 hr before harvested for in vivo ubiquitination assay. Bound proteins and inputs were detected by IB using antibodies as indicated. (**G**) NGFR enhancesMDM2-mediated p53 proteasomal degradation. H1299 cells were transfected with combinations of plasmids encoding p53, HA-MDM2 or RFP-NGFR followed by IB using antibodies as indicated. MG132 was supplemented to the medium for 6 hr.

The following figure supplement is available for figure 3:

**Figure supplement 1.** Knockdown of NGFR does not affect PUMA expression in the p53-null HCT116 cells.

lines (*Figure 3B*), whereas did not affect PUMA expression in HCT116 $^{p53-/-}$ cells (*Figure 3—figure supplement 1*). Consistently, knockdown of NGFR led to the global expression of a group of known p53 target genes, as measured by RNA-seq analysis in H460 cells (*Figure 3C*). This result was further confirmed by utilizing a second siRNA (siNGFR#2), as NGFR knockdown by this siRNA also induced the protein level of p53 and the mRNA levels of its target genes, including BAX, BTG2, MDM2, p21, and PUMA (*Figure 3D*). Also, ablation of NGFR prolonged p53's half-life (*Figure 3E*), suggesting that NGFR might regulate the stability of p53. Next, we tested if NGFR affects MDM2-induced p53 ubiquitination by conducting a p53 ubiquitination assay in H1299 cells with ectopic proteins. Surprisingly, ectopic NGFR enhanced MDM2-mediated p53 ubiquitination in a dose-dependent manner (*Figure 3F*), although this protein was found to mainly reside in the cytoplasmic membrane (*Barker, 2004*). Consistently, NGFR also enhanced MDM2-mediated p53 proteasomal degradation, which was abolished by the proteasome inhibitor MG132 (*Figure 3G*). Without MDM2, NGFR was unable to alter p53 protein level in cells (*Figure 4*). Altogether, these results indicate that NGFR reduces p53 stability by enhancing its MDM2-mediated ubiquitination and degradation.

## NGFR regulates p53 stability by binding to MDM2

Next, we tested if NGFR interacts with MDM2 by conducting a set of reciprocal co-immunoprecipitation (co-IP) assays. Indeed, ectopic NGFR bound to ectopic MDM2, and vice versa (*Figure 5A*). We then mapped their binding domains by performing a set of co-IP and GST-pull down assays. NGFR was specifically pulled down with the C-terminal aa 284–491 fragment, but not N-terminal fragments, of MDM2, which encompasses the zinc-finger, acidic, and ring-finger domains, in H1299 cells (*Figure 5C*). This result was verified by in vitro GST-pull down assays (*Figure 5D*). Using the same approach, we also mapped the MDM2-binding domain of NGFR. Unexpectedly, MDM2 interacted with the N-terminal extracellular and transmembrane domain, but not the C-terminal cytoplasmic domain, of NGFR (*Figure 5E*). Also, we validated the interaction between endogenous NGFR and MDM2 in human neuroblastoma SK-N-SH cells that sustains high level of NGFR (*Figure 5H*). To determine if this interaction takes place in the nucleus, we used H460 cells stably expressing NGFR, HCT116 $^{p53+/+}$ and SK-MEL-147 cells, fractionated them, and conducted IB assays. Indeed, full-length NGFR was present in both cytoplasm and nucleus (*Figure 5—figure supplement 1A–C*). Consistently, confocal microscopy analysis revealed that NGFR is present in all of the cellular compartments of SK-MEL-147 cells, albeit with less intensity in the nucleus than that in the cytoplasm (*Figure 5—figure supplement 1D*). Of note, the nuclear and cytoplasmic levels of NGFR increased upon Dox treatment likely due to p53 activation (*Figure 5—figure supplement 1D*). Also, NGFR was co-immunoprecipitated with MDM2 from the nuclear extracts of NGFR-stably expressed H460 (*Figure 5I*) and SK-MEL-147 cells in a reciprocal pattern (*Figure 5J*). Hence, these results demonstrate that the N-terminus of NGFR binds to the C-terminus of MDM2, and this interaction occurs in the nucleus.

## NGFR interacts with p53

While testing the endogenous NGFR-MDM2 binding, we also detected p53 in the NGFR-MDM2 complex (*Figure 5H–K*). Thus, we tested if p53 binds to NGFR directly or indirectly through MDM2. First, we conducted co-IP-IB and GST-Pull down assays. Indeed, NGFR and p53 co-immunoprecipitated with each other in reciprocal co-IP-IB assays (*Figure 6A–B*). Further confirming this interaction was the mapping of the NGFR-binding domain to the central DNA-binding domain of p53 using a similar co-IP-IB assay by transfecting plasmids encoding individual p53 fragments together with NGFR plasmid into H1299 cells (*Figure 6C*). This mapping was validated by a GST-pull down assay using purified GST-p53 fusion proteins with different p53 fragments. NGFR specifically bound to the full-length p53 and its central DNA-binding domain (aa 101–300) (*Figure 6D*). This finding explained why NGFR, MDM2 and p53 could form a ternary complex (*Figure 5H–k*). We also mapped the p53-binding domain of NGFR by conducting a set of co-IP-IB assays. Like MDM2, p53 also bound to the N-terminal region of NGFR (*Figure 6E*). Together with the result showing that NGFR binds to p53 in the nucleus (*Figure 5I–K*), these results demonstrate that the N-terminus of NGFR binds to the central DNA-binding domain of p53, and this interaction also occurs in the nucleus.

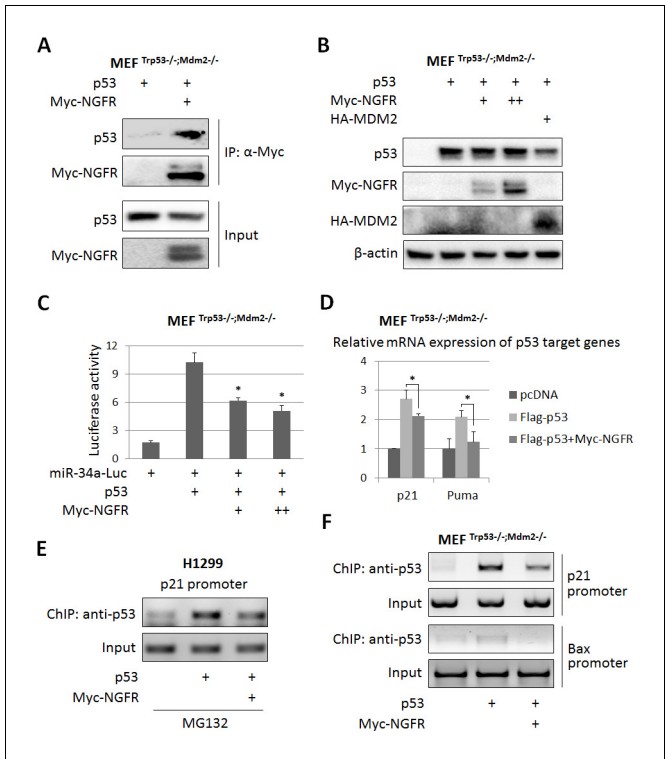

**Figure 4.** NGFR inactivates p53 independently of MDM2. (**A**) NGFR interacts with p53 in the absence of MDM2. MEF [p53-/-;Mdm2-/-] cells were transfected with plasmids encoding Myc-NGFR or p53 followed by co-IP-IB assays using antibodies as indicated. (**B**) NGFR does not affect p53 protein expression in the absence of MDM2. MEF [p53-/-;Mdm2-/-] cells were transfected with combinations of plasmids encoding Myc-NGFR, HA-MDM2 or p53 followed by IB using antibodies as indicated. (**C**) NGFR represses p53-induced luciferase activity independently of MDM2. MEF [p53-/-;Mdm2-/-] cells were transfected with plasmids as indicated in the figure and luciferase assay was performed as described in Materials and methods (Mean ± SEM, n = 3). Three biological replicates were used for p value, p*<0.05. (**D**) NGFR inhibits p53-induced target gene expression independently of MDM2. MEF [p53-/-; Mdm2-/-] cells were transfected with plasmids as indicated in the figure followed by q-PCR analysis (Mean ± SD, n = 3). Three technical replicates were used for p value, p*<0.05. (**E**) NGFR impedes p53 association with the p21 promoter in H1299 cells. Cells were transfected with plasmids as indicated and treated with MG132 for 6 hr before harvested for ChIP-PCR analyses. (**F**) NGFR impedes p53 association with the p21 and Bax promoters independently of MDM2. MEF [p53-/-;Mdm2-/-] cells were transfected with plasmids as indicated followed by ChIP-PCR analyses.

The following figure supplement is available for figure 4:

**Figure supplement 1.** MDMX, though binds to NGFR, is not required for NGFR-mediated p53 inactivation.

## NGFR attenuates p53 activity independently of MDM2

The finding that NGFR binds to the central DNA-binding domain of p53 suggests that this protein might directly regulate p53 transcriptional activity independently of MDM2. To test this idea, we first determined if NGFR can bind to p53 in the absence of MDM2 by introducing their expression plasmids alone or together into MEF[Mdm2-/-;p53-/-] cells followed by co-IP-IB assays. Indeed, NGFR was co-immunoprecipitated with p53 in these cells (*Figure 4A*). Interestingly and as also mentioned above, ectopic NGFR at different doses failed to decrease the level of ectopic p53 in MEF[Mdm2-/-;p53-/-] cells (*Figure 4B*), suggesting that NGFR cannot directly degrade p53 without MDM2. Then, we tested if NGFR could regulate p53 transcriptional activity in the absence of MDM2 by performing luciferase reporter assays. Interestingly, ectopic NGFR inhibited p53-induced luciferase activity driven by the miR-34a promoter in MEF[Mdm2-/-;p53-/-] cells in a dose dependent fashion (*Figure 4C*). Also, NGFR repressed ectopic p53-induced expression of p21 and Puma in MEF[Mdm2-/-;p53-/-] cells (*Figure 4D*).

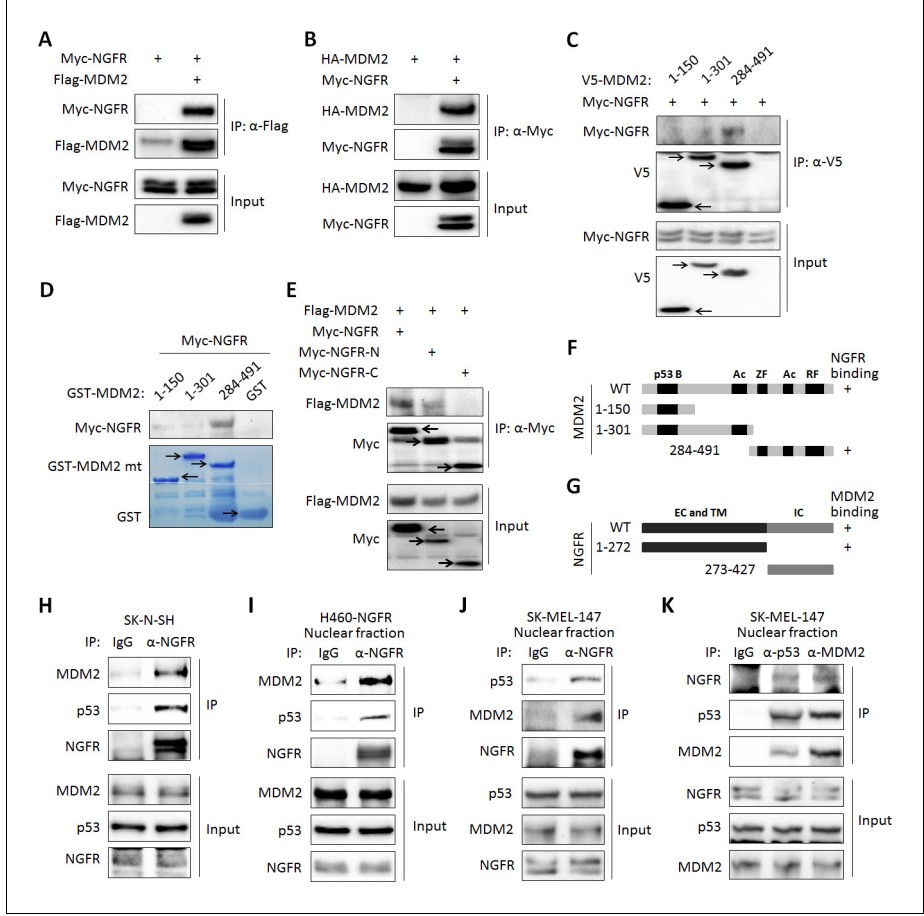

**Figure 5.** NGFR interacts with MDM2 in the nucleus. (**A,B**) NGFR interacts with MDM2. H1299 cells were transfected with plasmids encoding Myc-NGFR, Flag-MDM2 or HA-MDM2 followed by co-IP-IB assays using antibodies as indicated. (**C**) Mapping the NGFR binding domain on MDM2 by co-IP-IB assays. H1299 cells were transfected with the plasmid encoding V5-tagged MDM2 fragment, aa 1–150, aa 1–301 or aa 284–491, along with the Myc-NGFR-encoded plasmid. Co-IP-IB assays were performed using antibodies as indicated. (**D**) Mapping the NGFR binding domain on MDM2 by GST-pull down assay. The prokaryotic expressed GST-tagged MDM2 fragment, aa 1–150, aa 1–301 or aa 284–491, or GST protein alone was incubated with cell lysates overexpressing Myc-NGFR. Bound proteins were detected by IB using anti-NGFR or coomassie staining. (**E**) Mapping the MDM2 binding domain on NGFR. H1299 cells were transfected with the plasmid encoding Myc-tagged NGFR fragment, aa 1–272 or aa 273–427, along with the Flag-MDM2-encoded plasmid. Co-IP-IB assays were performed using antibodies as indicated. (**F**) A schematic of NGFR binding region on MDM2. (**G**) A schematic of MDM2 binding region on NGFR. (**H**) Endogenous interaction of NGFR and MDM2 in SK-N-SH cells. Cells treated with Doxorubicin for 12 hr and MG132 for 6 hr were harvested for co-IP-IB assays using antibodies as indicated. (**I,J,K**) NGFR interacts with both MDM2 and p53 in the nucleus. Nuclear fractions from NGFR-stably expressed H460 (**I**) and SK-MEL-147 cells (**J,K**) were subjected to co-IP-IB assays using antibodies as indicated.

The following figure supplement is available for figure 5:

**Figure supplement 1.** Full-length of NGFR localizes in both nucleus and cytoplasm.

Consistently, ectopic NGFR markedly reduced the association of p53 with the endogenous p21 and Bax promoters in these cells as analyzed by ChIP assays in H1299 or MEF[Mdm2-/-;p53-/-] cells (*Figure 4E*). Since MDMX, the partner of MDM2, also bound to NGFR (*Figure 4—figure supplement 1A*), MEF[Mdm2-/-; Mdmx-/-; p53-/-] cells were used to determine if MDMX is required for p53 inactivation by NGFR. NGFR still inhibited p53 activity as measured by p21 and Puma expression (*Figure 4—figure supplement 1B*). Collectively, these results demonstrate that NGFR also inhibits

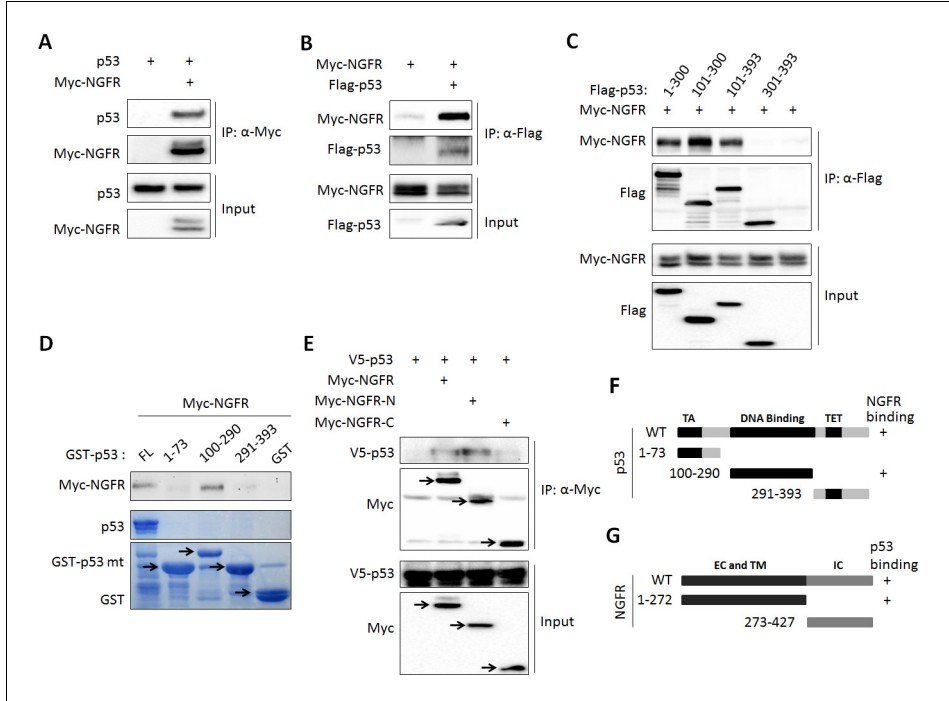

**Figure 6.** NGFR interacts with p53 in the nucleus. (**A**, **B**) NGFR interacts with p53. H1299 cells were transfected with plasmids encoding Myc-NGFR, Flag-p53 or p53 followed by co-IP-IB assays using antibodies as indicated. (**C**) Mapping the NGFR binding domain on p53 by co-IP-IB assays. H1299 cells were transfected with the plasmid encoding flag-tagged p53 fragment, aa 1–300, aa 101–300, aa 101–393 or aa 301–393, along with the Myc-NGFR-encoded plasmid. Co-IP-IB assays were performed using antibodies as indicated. (**D**) Mapping the NGFR binding domain on p53 by GST-pull down assay. The prokaryotic expressed GST-tagged full-length p53 or p53 fragment, aa 1–73, aa 100–290 or aa 291–393, or GST protein alone was incubated with cell lysates overexpressing Myc-NGFR. Bound proteins were detected by IB using anti-NGFR or coomassie staining. (**E**) Mapping the p53-binding domain on NGFR. H1299 cells were transfected with the plasmid encoding Myc-tagged NGFR fragment, aa 1–272 or aa 273–427, along with the V5-p53-encoded plasmid. Co-IP-IB assays were performed using antibodies as indicated. (**F**) A schematic of NGFR binding region on p53. (**G**) A schematic of p53 binding region on NGFR.

p53 by directly binding to its central DNA-binding domain and preventing its association with its target promoter.

## Depletion of NGFR sensitizes tumor cells to cisplatin and doxorubicin

The discovery that NGFR is an inhibitor of p53 in cancer cells suggested the possibility of that depletion of this protein could sensitize cancer cells to chemotherapeutic agents. We tested this idea by treating H460 and SK-N-SH cells with different doses of Dox and Cisplatin after introducing control or NGFR siRNAs into the cells. Indeed, both of the chemotherapeutic drugs induced p53 and its target PUMA or p21 more apparently when the cells were transfected with NGFR siRNA compared to those with control siRNA in a dose-dependent fashion (*Figure 7A–D*). Also, this more robust p53 induction led to more cytotoxicity, as the IC50 of Cisplatin in NGFR siRNA-transfected H460 was 2.2-fold less than that in control siRNA-transfected H460 cells (*Figure 7E*). The result was repeated in HCT116 [p53+/+] cells, in which the IC50 of Cisplatin dropped by 1.6 folds in response to NGFR knockdown (*Figure 7G*). This enhanced cytotoxicity was p53-dependent, as less or no significant difference in cell viability was observed when p53-null H1299 or HCT116 [p53-/-] cells were used for the same experiment (*Figure 7F and H*). These results suggest that NGFR confers resistance of cancer cells to chemotherapeutic-triggered p53 activation and cytotoxicity, and thus, inactivation of NGFR can sensitize the cells to chemotherapy.

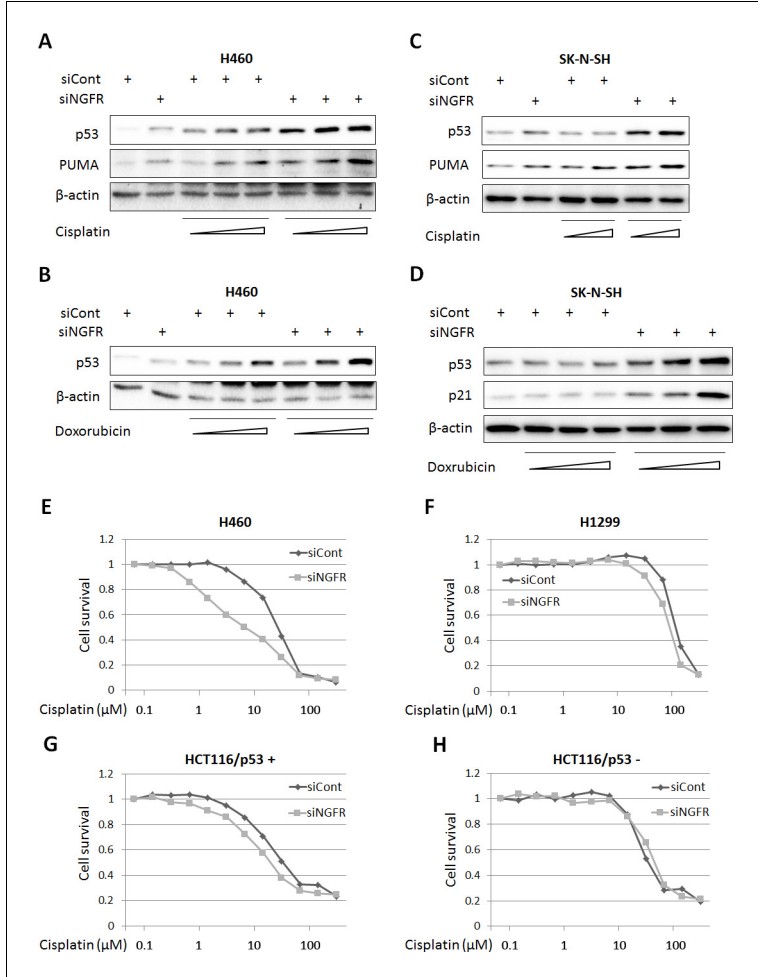

**Figure 7.** NGFR confers cancer cells resistance to chemotherapeutic agents. (**A,B**) NGFR knockdown enhances Cisplatin or Doxorubicin-triggered p53 activation in H460 cells. Cells were transfected with NGFR or control siRNA for 72 hr and Cisplatin (**A**) or Doxorubicin (**B**) was supplemented in the medium 12 hr before the cells were harvested for IB using antibodies as indicated. (**C,D**) NGFR knockdown enhances Cisplatin (**C**) or Doxorubicin (**D**) - triggered p53 activation in SK-N-SH cells.The same experiments as shown in (**A,B**) were performed except that the SK-N-SH cell line was used instead. (**E,F**) NGFR knockdown sensitizes H460 (**E**) but not H1299 cells (**F**) to Cisplatin treatment. Cells were transfected with NGFR or control siRNA and seeded in 96-well plates the next day. Cisplatin was supplemented 48 hr before cell viability was determined using CCK-8 as described in Materials and methods. (**G,H**) NGFR knockdown sensitizes HCT116 [p53+/+] (**G**) but not HCT116 [p53-/-] cells (**H**) to Cisplatin treatment. The same experiments described in (**E,F**) were performed except for using different cell lines.

## Depletion of NGFR suppresses human xenograft tumor growth

To further translate the above findings into biological significance, we then determined if NGFR is required for tumor growth in mice by generating a xenograft tumor model using severe combined immunodeficiency (SCID) mice and H460 cells. NGFR shRNA or control shRNA-packaged lentivirus was produced to infect H460 and SK-MEL-147 cells. As expected (*Figure 3B*), the expression of p53 and its target genes p21, PUMA, and MDM2 was induced by knocking down NGFR using the shRNA-expressing lentiviruses (*Figure 8A* and *Figure 8—figure supplement 1A*). Correspondingly, H460 cells with deficient NGFR grew slowly (*Figure 8B*) and formed less colonies (*Figure 8C*). In line with this result, the xenograft tumors derived from NGFR-shRNA H460 cells grew remarkably slower than those from control-shRNAH460 cells in the SCID mice (*Figure 8D*). Also, the average tumor weight from control cells was ~ten-fold heavier than that from NGFR-shRNA cells by the end of the

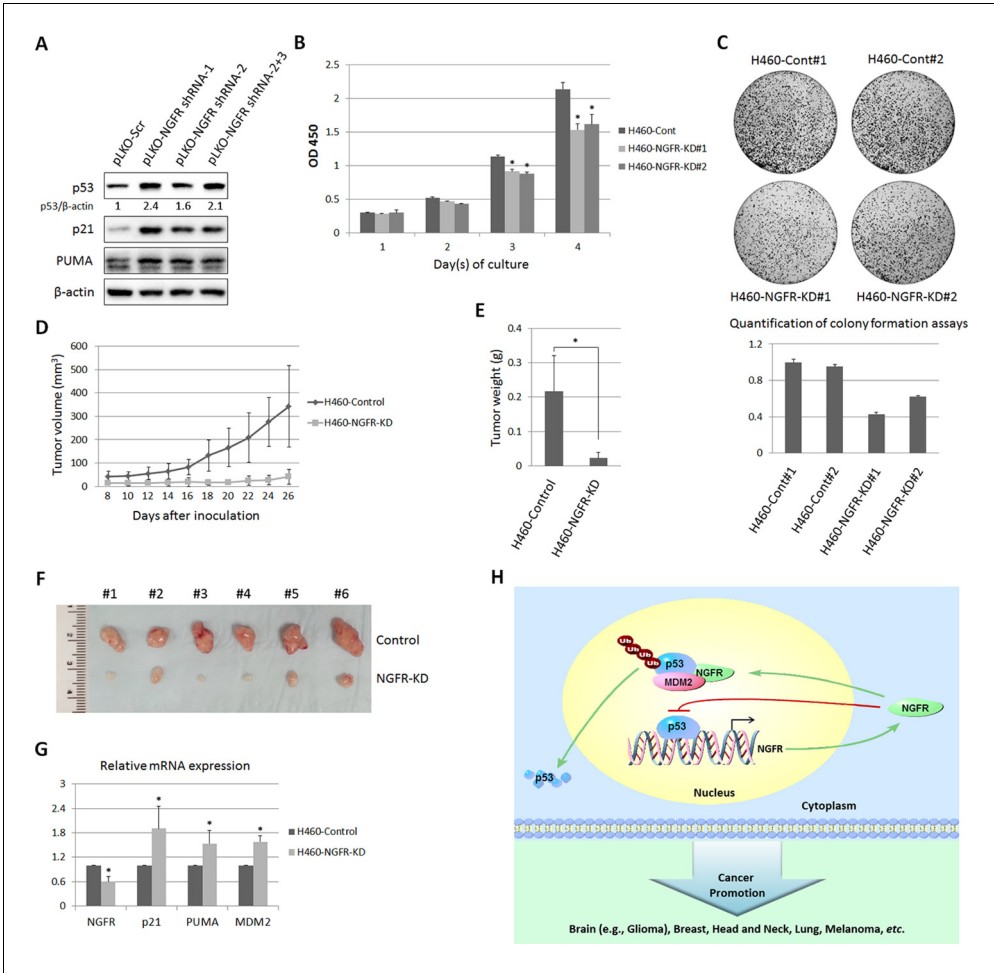

**Figure 8.** NGFR is required for tumor growth in vivo. (**A**) Lentiviral-based knockdown of NGFR induces p53 pathway. H460 cells were transduced with lentivirus expressing NGFR or control shRNA for 72 hr followed by IB using antibodies as indicated. (**B**) H460 cells stably expressing NGFR shRNA show suppressed cell survival. H460 cells stably expressing NGFR or control shRNA were seeded in 96-well plates and cell viability was evaluated every 24 hr (Mean ± SEM, n = 3). Three biological replicates were used for p value, p*<0.01. (**C**) H460 cells stably expressing NGFR shRNA exhibit restrained clonogenic capacity. H460 cells stably expressing NGFR or control shRNA were seeded on 10-cm plates and colonies were fixed by methnol and stained with crystal violet solution (upper panel). Quantification of colonies is shown in the lower panel. (**D**) H460 cells stably expressing NGFR shRNA reveal less xenograft tumor volume in average (Mean ± SEM, n = 6). (**E**) H460 cells stably expressing NGFR shRNA reveal less xenograft tumor weight in average (Mean ± SEM, n = 6). Six pairs of tumors were used for p value, p*<0.01. (**F**) Representative xenograft tumors reveal that NGFR knockdown dramatically suppressed tumor growth in vivo. (**G**) The p53 pathway is activated by NGFR knockdown in the xenograft tumors. The expression of p21, PUMA and MDM2 was determined by q-PCR (Mean ± SEM, n = 3). Three pairs of tumors were used for p value, p*<0.01. (**H**) A model for NGFR regulation of the MDM2-p53 loop in cancer. NGFR inactivates p53 through two mechanisms: 1) by directly associating with MDM2 and enhancing MDM2-mediated p53 ubiquitination and proteasomal degradation, and 2) by repressing p53 transcriptional activity through direct interaction with its DNA binding domain, consequently leading to tumorigenesis.

The following figure supplements are available for figure 8:

**Figure supplement 1.** Lentivirus-mediated knockdown of NGFR activates p53 in a melanoma cell line and suppresses in vivo tumor growth.

**Figure supplement 2.** NGFR is apt to be amplified in breast cancers sustaining wild-type p53.

experiment (*Figure 8E*). Tumors derived from NGFR-shRNA H460 cells are much smaller than those derived from control-shRNA H460 cells (*Figure 8F* and *Figure 8—figure supplement 1B*). Because of the diminutive size of the tumors generated from the NGFR knockdown group, we were unable to collect sufficient amounts of tumor tissues for WB and IHC staining. But, by performing q-PCR analysis, we found that the expression of p21, PUMA, and MDM2 is significantly elevated in response to NGFR depletion in the xenograft tumors (*Figure 8G*), consistent with the in vitro results (*Figure 3C and D*). Thus, these results demonstrate that NGFR is required for tumor growth in vivo.

## Discussion

Regardless of the crucial importance of p53 in preventing tumorigenesis, there are still about 50% of all types of human cancers that harbor wt p53. Here, we identified NGFR as a novel p53 inactivator that serves as an example of a previously unidentified oncoprotein that may allow tumorigenesis in lieu of wt p53. NGFR inhibited p53 transcriptional activity as a feedback regulator in the nucleus of several cancer cells (*Figure 8H*), even though it normally resides in the cellular membrane of a neuron in both the central and peripheral nervous systems (*Barker, 2004*). Remarkably, targeting NGFR led to the suppression of tumor cell growth in vitro and in vivo (*Figures 2*, *7* and *8*). Also, NGFR was highly expressed in human gliomas (*Figure 2H and I* and *Figure 2—figure supplement 1B*). In addition to the xenograft model generated by human lung non-small cell carcinoma H460 cells (*Figure 8*), NGFR has also been shown by others to enhance cell survival of schwannoma (*Ahmad et al., 2014*; *Gentry et al., 2000*), melanoma (*Boiko et al., 2010*), and breast cancer (*Descamps et al., 2001*), and to promote metastasis of glioma (*Johnston et al., 2007*), melanoma (*Boiko et al., 2010*), and head and neck cancers (*Murillo-Sauca et al., 2014*) (*Figure 8H*). Most of these cancers harbor wt p53 (*Soussi et al., 2005* and IRAC TP53 database, R17). Analysis of the TCGA genomic database also showed that the NGFR gene is often amplified in human breast cancers that display no p53 mutation (*Cerami et al., 2012*; *Gao et al., 2013*) (*Figure 8—figure supplement 2*). Hence, our study highly suggests that NGFR plays an oncogenic role by targeting p53 in these human cancers that harbor wt p53. Also, our study leads to several new findings to the p53 field.

### NGFR feedback negation of p53 functions

First, we demonstrate that *NGFR* is a bona fide p53 target gene. The expression of *NGFR* at both mRNA and protein levels was induced by INZ, Dox, and 5-FU in several p53-positive, but not in p53-null, cancer cells (*Figure 1A–E*). Also, overexpression of wt, but not mutant, p53 stimulated the expression of NGFR in p53-null cancer cells (*Figure 1F and G*). In addition, knockdown of p53 in HCT116 $^{p53+/+}$ and H460 impaired the induction of NGFR mRNA by Dox and 5-FU (*Figure 1H and I*). Furthermore, p53 stimulated the expression of luciferase gene driven by the p53RE sequence derived from the NGFR promoter (*Figure 1J–L*). Finally, in response to Dox, p53 was induced to associate with the p53RE-containing NGFR promoter in H460 cells. Undoubtedly, *NGFR* is an authentic p53 target gene, which is also in line with a recent finding that NGFR is a transcriptional target for the p53 analog, p73, in the mouse nerve system (*Niklison-Chirou et al., 2013*).

However, NGFR surprisingly counteracted p53 function, as knockdown of this protein in p53-containing HCT116 or H460 cells led to drastic apoptosis and significant growth arrest (*Figure 2A–F*). Also, knockdown of NGFR led to a more significant decrease of colony formation in p53-containing H460 cells compared to p53-deficient H1299 cells (*Figure 2G*). Of note, the suppressive effect of NGFR knockdown on H1299 colony formation might be due to the oncogenic function of this protein via other oncoproteins, such as NF-κB (*Carter et al., 1996*). But, this result does not affect our conclusion that negation of p53 function is one of the oncogenic functions of NGFR, as it is strongly supported by the evidence that knockdown of NGFR leads to activation of the p53 pathway in H460, HepG2, SK-N-SH, HCT116 $^{p53+/+}$ and melanoma SK-MEL cells (*Figure 3B–D*). Remarkably, a number of p53 responsive target genes were induced to different degrees when NGFR was knocked down in H460 cells (*Figure 3C*). The NGFR inhibition of p53 is further supported by the fact that it inactivates p53 by two distinct mechanisms as detailed below.

### NGFR assists MDM2 in p53 degradation

One mechanism for NGFR inactivation of p53 is by working with MDM2. First, ectopic NGFR enhanced MDM2-mediated ubiquitination and proteosomal degradation of p53 (*Figure 3F and G*).

Inversely, knockdown of NGFR led to p53 stabilization (*Figure 3E*). These actions appeared to be implemented through direct interaction of NGFR with MDM2, as NGFR interacted with MDM2 both in vitro (*Figure 5D*) and in the nucleus (*Figure 5I–K*) via its N-terminal region (*Figure 5E*). Although stress signals (*Palmero et al., 1998*; *Shieh et al., 1997*; *Tang et al., 2008*; *Zhang et al., 1998*; *Zindy et al., 1998*) (*Zhang and Lu, 2009*; *Zhou et al., 2012*, *2015a*) have been shown to compromise MDM2-induced ubiquitination and proteolysis of p53 and thus activate p53 in normal cells, cancer cells employ oncogenic molecules, such as MDMX (*Gembarska et al., 2012*; *Lam et al., 2010*; *Slack et al., 2005*; *Wade et al., 2013*), which strengthen MDM2-mediated inhibition of p53 and thus induce cell resistance to p53 activation. Our data indicates that NGFR is one of these molecules as knockdown of NGFR enhances chemotherapeutic agents-induced p53 activation (*Figure 7A–D*) and cell growth inhibition (*Figure 7E– H*).

## NGFR inactivates p53 independent of MDM2

Surprisingly, NGFR can also inactivate p53 independent of MDM2. First, NGFR binds to p53 in cells and in vitro (*Figure 6A–E*). Interestingly, NGFR bound to the central DNA-binding domain of p53 (*Figure 6D*), and reduced the binding of p53 to the p21 and BAX promoters (*Figure 4E–F*). All of these activities of NGFR are MDM2-independent, as they were all detected in MEF$^{p53-/-/MDM2-/-}$ cells (*Figure 4A–6D*). However, without MDM2, ectopic NGFR was unable to mediate p53 degradation (*Figure 4B*). These results demonstrate that NGFR can directly bind to the central DNA-binding domain of p53 and impair its ability to bind to its target promoters in addition to facilitating MDM2-dependent p53 ubqiutination and degradation. Hence, our study illustrates two novel mechanisms by which NGFR inactivates p53, which are ligand-independent as further described below.

## NGFR inactivates p53 in the nucleus independently of ligand

Our findings that NGFR can directly and indirectly suppress p53 activity were unexpected, as NGFR was originally found to be a transmembrane pan-receptor involved in the initiation, development, and maintenance of the nervous system and human cancers (*Lee et al., 2001*; *Molloy et al., 2011*; *Patapoutian and Reichardt, 2001*), which binds with low affinity to all mature neurotrophins, including nerve growth factor (NGF), brain-derived neutrophic factor (BDGF), neurotrophin 3 (NT-3), and neurotrophin 4/5 (NT-4/5), as well as their precursors, pro-neurotrophins, with high affinity (*Barker, 2004*). Although a previous study showed that in response to neurotrophin signaling, NGFR might mediate p53-dependent neuron cell death by activating the JNK pathway (*Aloyz et al., 1998*), our studies as presented here unveil a novel nuclear function of this membrane receptor, i.e., to promote cancer cell proliferation and survival by directly inhibiting p53 transcriptional activity and indirectly destabilizing p53 protein via MDM2. Convincingly, we showed that NGFR binds to MDM2 in the nucleus and assists this E3 ligase to ubiquitinate p53 and promote its degradation (*Figures 3*,*5*). Also, independently of MDM2, NGFR directly bound to the central domain of p53 in the nucleus and prevented p53 from binding to its target promoters (*Figures 4*,*6*). These inhibitory activities of NGFR toward p53 are ligand-independent, as they occurred in the nucleus and NGF treatment did not appear to affect p53 level and activity in H460, HCT116, and HepG2 cancer cells tested (Data not shown). Hence, our results for the first time uncover a ligand-independent and nuclear p53 inhibitory function of this membrane neurotrophic receptor by acting on the MDM2-p53 loop.

## Cancer cells hijack NGFR to inactivate p53

Although it remains to be studied if NGFR can also regulate p53 stability and activity in a negative feedback fashion in normal nerve systems, our findings demonstrate that cancer cells hijack this anti-p53 activity of NGFR toward their growth advantage in a way similar to that for MDM2 and MDMX inactivation of p53 in cancers (*Momand et al., 1992*; *Oliner et al., 1992*, *1993*; *Shvarts et al., 1996*; *Wu et al., 1993*), as knockdown of NGFR markedly induced p53-dependent apoptosis and cytotoxicity as well as eliminated xenograft tumor growth (*Figures 2–8*). In line with our findings, a growing body of evidence has identified NGFR as a robust cell surface biomarker not only for neural crest stem cells, but also cancer initiating or stem-like cells (*Tomellini et al., 2014*). Also, NGFR was highly expressed in a number of cancers, including the cancer initiating cells of melanoma (*Boiko et al., 2010*), squamous cell carcinomas (*Murillo-Sauca et al., 2014*), osteosarcoma

(*Tian et al., 2014*), brain cancer (*Biagiotti et al., 2006*), breast cancer (*Kim et al., 2012*), and neuroblastoma (*Biagiotti et al., 2006*). Particularly, patient derived NGFR-positive, but not NGFR-negative, melanoma cells were remarkably capable of generating tumors, promoting metastasis, and maintaining self-renewal (*Boiko et al., 2010*). Also, we detected the high expression of NGFR in several melanoma cell lines (*Figure 3B*) and primary human gliomas (*Figure 2H,I* and *Figure 2—figure supplement 1B*). Several of these cancers, such as breast cancer, melanoma, osteocarcoma, and neuroblastoma, often display less or no TP53 mutations (*Petitjean et al., 2007*; *Soussi et al., 2005*). Although MDM2 and MDMX have been shown to play a role in suppressing p53 in these cancers (*Gembarska et al., 2012*; *Lam et al., 2010*; *Slack et al., 2005*; *Wade et al., 2013*), it is highly likely that NGFR might also contribute to the reason why these cancers harbor wt p53. Our findings as demonstrated here further support this likelihood and provide novel mechanisms underlying NGFR inactivation of p53 in these cancers or cancer stem cells. Together with the aforementioned studies (*Boiko et al., 2010*; *Tomellini et al., 2014*), our findings also suggest that NGFR might play a role in maintaining the renewal and proliferating capabilities of stem cells or cancer stem cells by inactivating p53, as p53 has been shown to be crucial for stem cell differentiation and apoptosis (*Krizhanovsky and Lowe, 2009*) and to be a major roadblock for these stem cells to renew and proliferate (*Cicalese et al., 2009*; *Hong et al., 2009*; *Kawamura et al., 2009*; *Marion et al., 2009*; *Utikal et al., 2009*).

In contrast to our findings and the oncogenic role as described above, NGFR has also been shown to exert a tumor suppressive function by repressing tumor growth in several cancers (*Molloy et al., 2011*), which has been largely attributed to its receptor activity (*Barker, 2004*; *Bredesen and Rabizadeh, 1997*; *Carter et al., 1996*; *El Yazidi-Belkoura et al., 2003*; *Khursigara et al., 2001*; *Nykjaer et al., 2004*; *Tomellini et al., 2014*). One previous study showed that the pro-neurotrophin-NGFR signaling can induce neuron death through the JNK-p53 pathway (*Aloyz et al., 1998*), but little is known about if this is also true in cancer or if NGFR can activate p53 independent of its ligands. All these studies did not notice the receptor-independent intracellular function of NGFR. Thus, our study not only offers new insights into the mechanism of intracellular NGFR-mediated tumor promotion, but also demonstrates that this oncogenic function is p53-dependent, which could explain why several human cancers that express high levels of NGFR harbor wt p53. Finally, our study suggests NGFR as a potential therapeutic target for cancers that harbor wt p53 and high levels of NGFR.

# Materials and methods

## Plasmids and antibodies

The RFP-tagged plasmid pDSRed-NGFR was purchased from Addgene created by Dr. Moses Chao. The Myc-tagged NGFR expression plasmid was generated by inserting the full-length cDNA amplified by PCR from pDSRed-NGFR into the pcDNA3.1/Myc-His vector, using the following primers, 5-CCGGAATTCATGGGGGCAGGTGCCACC-3 and 5-CGCGGATCCCACCGGGGATGTGGCAGT-3. The Myc-tagged plasmids expressing NGFR fragment, aa 1–272 or aa 273–427, were generated by the same approach using the corresponding primers. The pGL3-RE1 and RE2 plasmids were generated by inserting the genomic DNA covering p53 RE1 or RE2 into the pGL3-promoter vector using the following primers, 5-CGGGGTACCTTCTACTGTCATGTCAAAGGAA-3 and 5-CCGCTCGAGCCCTCCAGCTACTACTCAGAC-3 for RE1; 5-CGGGGTACCGGCAAGTGGCATTGGTGGTA-3 and 5-CCGCTCGAGTCGTTTGTAAAGTGGGCATAA-3 for RE2. The lentiviral-based plasmid NGFR shRNA-1 was generated by inserting the following sequence, 5-CCGGCCGAGCACATAGACTCCTTTACTCGAGTAAAGGAGTCTATGTGCTCGGTTTTTG-3 into pLKO.1 vector. The plasmids NGFR shRNA-2 and −3 were purchased (Sigma-Aldrich, St. Louis, MO, USA). The plasmids encoding HA-MDM2, Flag-MDM2, V5-MDM2 fragments, GST-MDM2 fragments, HA-MDMX, p53, Flag-p53, His-Ub and pGL4-miR-34a-luciferase were described previously (*Dai and Lu, 2004*; *Dai et al., 2004*; *Zhou et al., 2015b*). Plasmids encoding GST-tagged p53 fragments and Flag-tagged p53 fragments were gifts from Wei Gu and Mushui Dai, respectively. V5-tagged p53 was purchased from Addgene (*Junk et al., 2008*). Anti-Flag (Sigma-Aldrich, St. Louis, MO, USA), anti-Myc (9E10, Santa Cruz Biotechnology, Santa Cruz, CA, USA), anti-V5 (E10, Thermo Scientific, Waltham, MA, USA), anti-GFP (B-2, Santa Cruz Biotechnology), anti-NGFR (Millipore, Billerica, MA, USA; EP1039Y, GeneTex, Irvine,

CA, USA and D4B3, Cell signaling Technology, Danvers, MA, USA), anti-p53 (DO-1, Santa Cruz Biotechnology), anti-p21(CP74, Neomarkers, Fremont, CA, USA), anti-PUMA (H-136, Santa Cruz Biotechnology) and anti-β-actin (C4, Santa Cruz Biotechnology) were commercially purchased. Antibodies against MDM2 (2A9, 2A10 and 4B11) were previously described (*Dai and Lu, 2004*; *Dai et al., 2004*).

## Cell culture and transient transfection

Human cancer cell lines H460, H1299, HCT116$^{p53+/+}$, HCT116$^{p53-/-}$, HepG2, PCL/PRF/5, U2OS, SK-MEL-103, SK-MEL-147, MEF$^{p53-/-;Mdm2-/-}$ and MEF$^{p53-/-;Mdm2-/-; Mdmx-/-}$ cells were cultured in Dulbecco's modified Eagle's medium (DMEM) supplemented with 10% fetal bovine serum, 50 U/ml penicillin and 0.1 mg/ml streptomycin. The human neuroblastoma cell line SK-N-SH was cultured in RPMI 1640 medium supplemented with 10–15% fetal bovine serum, 50 U/ml penicillin and 0.1 mg/ml streptomycin. All cells were maintained at 37°C in a 5% $CO_2$ humidified atmosphere. Cells seeded on the plate overnight were transfected with plasmids as indicated in figure legends using TurboFect transfection reagent following the manufacturer's protocol (Thermo Scientific). Cells were harvested at 30–48 hr post-transfection for future experiments.

## GST fusion protein association assay

GST-tagged MDM2 fragments or p53 fragments were expressed in *E. coli* and conjugated with glutathione-Sepharose 4B beads (Sigma-Aldrich). Protein-protein interaction assays were conducted by using cell lysates with mammalian-expressed Myc-NGFR. Briefly, the cell lysates were incubated and gently rotated with the glutathione-Sepharose 4B beads containing 500 ng of GST-MDM2 fragments, GST-p53 fragments or GST only at 4°C for 4 hr. The mixtures were washed three times with GST lysis buffer (50 mM Tris/HCl pH 8.0, 0.5% NP-40, 1 mM EDTA, 150 mM NaCl, 10% glycerol). Bound proteins were analyzed by IB with the antibodies as indicated in the figure legends.

## Chromatin immunoprecipitation

Chromatin immunoprecipitation (ChIP) assay was performed using antibodies as indicated in the figure legends and described previously (*Liao and Lu, 2013*). The reverse cross-linked immuoprecipitated DNA fragments were purified using GeneJET gel extraction kit (Thermo Scientific) followed by PCR analyses for the p53-responsive DNA elements on the promoters of human *NGFR* and *p21* and mouse *p21* using the following primers, 5-GACTCCAACCTTGCTAATTCCT-3 and 5-TGACCTTCAC-CAGTTCTCACT-3 for human *NGFR*, 5-GCTCCCTCATGGGCAAACTCACT-3 and 5-TGGCTGGTCTACCTGGCTCCTCT-3 for human *p21*, and 5-CCTTTCTATCAGCCCCAGAGGATACC-3 and 5-GACCCCAAAATGACAAAGTGACAA-3 for mouse *p21*.

## Reverse transcription and quantitative PCR analyses

Total RNA was isolated from cells using Trizol (Invitrogen, Carlsbad, CA, USA) following the manufacturer's protocol. Total RNAs of 0.5 to 1 µg were used as templates for reverse transcription using poly-(T)20 primers and M-MLV reverse transcriptase (Promega, Madison, WI, USA). Quantitative PCR (qPCR) was conducted using SYBR Green Mix according to the manufacturer's protocol (BioRad, Hercules, CA, USA). The primers for human NGFR, mouse p21 and Puma are as follows, 5-CCTGGA-CAGCGTGACGTTC-3 and 5-CCCAGTCGTCTCATCCTGGT-3 for NGFR, 5-CCAGCAGAATAAAAGGTGCCACAGG-3 and 5-GCATCGCAATCACGGCGCAA-3 for mouse p21, and 5-ACGACCTCAACGCGCAGTACG-3 and 5-GAGGAGTCCCATGAAGAGATTG-3 for mouse Puma. The primers for human BTG2, BAX, MDM2, p21, PUMA and GAPDH were previously described (*Sun et al., 2010*; *Zhou et al., 2015b*).

## Flow cytometry analyses

Cells transfected with siRNAs as indicated in the figure legends were fixed with ethanol overnight and stained in 500 ml of propidium iodide (Sigma-Aldrich) stain buffer (50 µg/ml PI, 200 µg/ml RNase A, 0.1% Triton X-100 in phosphate-buffered saline) at 37°C for 30 min. The cells were then analyzed for DNA content using a BD Biosciences FACScan flow cytometer (BD Biosciences, San Jose, CA, USA). Data were analyzed using the CellQuest (BD Biosciences) and Modfit (Verity,

Topsham, ME, USA) software programs. Sub-G1 as an indicator of apoptosis was measured by determining the number of events in channels 0 to 40 in a 256 channel histogram.

### Cell viability assay

To assess the long term cell survival, the Cell Counting Kit-8 (CCK-8) (Dojindo Molecular Technologies, Rockville, MD, USA) was used according to the manufacturer's instructions. Cell suspensions were seeded at 2000 cells per well in 96-well culture plates at 12 hr post-transfection. Cell viability was determined by adding WST-8 at a final concentration of 10% to each well, and the absorbance of the samples was measured at 450 nm using a Microplate Reader (Molecular Device, SpecrtraMax M5e, Sunnyvale, CA, USA) every 24 hr for 4 days.

### RNA-sequencing and bioinformatics analysis

The RNA-sequencing service was provided by the Genomics and Biostatistics Core at the Tulane Center for Aging and the RNA-sequencing data were analyzed by the Cancer Crusaders Next Generation Sequence Analysis Core of the Tulane Cancer Center. Experiments were triplicate and genes with over 1.5-fold increase in expression (p<0.05) were shown in the study.

### Colony formation assay

Cells were trypsinized and seeded with the same amount on 10-cm plates following siRNA transfection for 12 to 18 hr. The medium was changed every 3 days until the colonies were visible. Puromycin was added in the medium when the stable cell lines were used in the experiment. Cells were then fixed by methonal and stained by crystal violet solution at RT for 30 min. ImageJ was used for quantification of the colonies.

### Immunobloting

Cells were harvested and lysed in lysis buffer consisting of 50 mM Tris/HCl (pH7.5), 0.5% Nonidet P-40 (NP-40), 1 mM EDTA, 150 mM NaCl, 1 mM dithiothreitol (DTT), 0.2 mM phenylmethylsulfonyl fluoride (PMSF), 10 µM pepstatin A and 1 mM leupeptin. Equal amounts of clear cell lysate (20–80 µg) were used for immunoblotting (IB) analyses as described previously (*Zhou et al., 2013*).

### Luciferase reporter assay

Cells were transfected with pCMV-β-galactoside together with the plasmids as indicated in the figures. Luciferase activity was determined and normalized by a factor of β-Gal activity in the same assay as described previously (*Liu et al., 2014*; *Zhou et al., 2015b*).

### In vivo ubiquitination assay

H1299 cells were transfected with plasmids encoding p53, HA-MDM2, His-Ub or Myc-NGFR as indicated in the figure legends. At 48 hr after transfection, cells were harvested and split into two aliquots, one for IB and the other for ubiquitination assays. In vivo ubiquitination assays were conducted as previously described (*Zhou et al., 2013*). Briefly, cell lysates were incubated with Ni-NTA beads that capture His-tagged proteins/complex at RT for 4 hr. The captured proteins were eluted and analyzed by IB with the indicated antibodies.

### Immunoprecipitation

Immunoprecipitation (IP) was conducted using antibodies as indicated in the figure legends and described previously (*Zhou et al., 2013*). Briefly, ~500 to 1000 µg of proteins were incubated with the indicated antibody at 4°C for 4 hr or overnight. Protein A or G beads (Santa Cruz Biotechnology) were then added and the mixture was left to incubate at 4°C for additional 1 to 2 hr. The beads were washed at least three times with lysis buffer. Bound proteins were detected by IB with antibodies as indicated in the figure legends.

### Immunostaining and confocal microscopy

Cells were fixed with methanol in −20°C for overnight. The fixed cells were washed by PBS and blocked with 8% BSA in PBS for 1 hr followed by incubation with primary antibodies (D4B3 for NGFR, 1:200 dilution; DO-1 for p53, 1:100 dilution) in 2% BSA in 4°C for overnight. The cells were

then washed and incubated with the corresponding secondary antibodies and DAPI. Images were acquired with a confocal microscope (Olympus FV1000).

## Human glioma specimens

Glioma and adjacent normal tissue samples were collected and archived at the First Affiliated Hospital of Nanchang University, Jiangxi, China. Due to the statistical consideration of small sample size ($\geq$40), we collected 48 pairs of Glioma and matched adjacent normal tissues. Fresh glioma tissues and paired noncancerous tissues were immediately snap-frozen in liquid nitrogen and stored at $-80°C$ until their use in immunoblotting. Immunohistochemical staining was performed as previously described (*Zeng et al., 2014*). All patients provided written informed consent to participate in the study and all primary glioma samples without preoperative radiotherapy were included and confirmed by pathologists.

## RNA interference

The siRNAs against NGFR and p53 (Life Technologies, Carlsbad, CA, USA) were commercially purchased. 40~60 nM of siRNAs were introduced into cells using TurboFect transfection reagent following the manufacturer's protocol. Cells were harvested ~72 hr after transfection for IB or qPCR.

## Generating stable cell lines

Lentiviral plasmids based on pLKO.1 were packaged with the 2nd Generation Packaging System. Briefly, pLKO.1 plasmids containing scrambled or NGFR shRNAs, along with the packaging plasmids pMD2.G and pCMV-dR8.2, were transfected into 293T cells. The cells were maintained at 37°C in a 5% $CO_2$ humidified atmosphere for 72 hr and the supernatant was harvested to infect H460 or SK-MEL-147 cells. The medium was changed by overnight infection and cells were split to two aliquots, one for IB analysis and the other for selection with 2 μg/ml puromycin.

## Mouse xenograft studies

Seven-week-old female NOD/SCID mice were purchased from Jackson Laboratories. Mice were subcutaneously inoculated with 3 x 10$^6$ H460 cells infected with lentivirus encoding control shRNA or NGFR shRNA in the right and left flanks, respectively. Tumor growth was monitored every other day with electronic digital calipers (Thermo Scientific) in two dimensions. Tumor volume was calculated with the formula: tumor volume (mm$^3$) = (length x width$^2$)/2 (*Zhang et al., 2012*). Mice were sacrificed by euthanasia and tumors were harvested and weighed. Mice with the largest or smallest tumor size were excluded from our study. To detect p53 activation and apoptotic signals in vivo, the tumors were disrupted in Trizol and subjected to RT-qPCR analyses.

## Acknowledgements

We thank Dr Malwina Czarny-Ratajczak for RNA seq analysis, who is supported by the COBRE grant (P20 GM103629), Mary Price for flow cytometry analysis, Wei Gu and Mu-shui Dai for offering plasmids, and Daniel Nguyen for proofreading. SL was supported in part by National Natural Science Foundation of China (31171359, 31460305, and 81472607). HL was supported in part by NIH-NCI grants R01CA095441, R01CA172468, R01CA127724, and R21CA190775 as well as the Reynolds and Ryan Families Chair Fund in Translational Cancer.

## Additional information

### Funding

| Funder | Grant reference number | Author |
| --- | --- | --- |
| National Natural Science Foundation of China | 31171359 | Shiwen Luo |
| National Natural Science Foundation of China | 31460305 | Shiwen Luo |

| National Natural Science Foundation of China | 81472607 | Shiwen Luo |
| NIH Office of the Director | CA095441 | Hua Lu |
| NIH Office of the Director | CA172468 | Hua Lu |
| NIH Office of the Director | CA127724-05 | Hua Lu |
| NIH Office of the Director | CA190775 | Hua Lu |

The funders had no role in study design, data collection and interpretation, or the decision to submit the work for publication.

## Author contributions

XZ, QH, Conception and design, Acquisition of data, Analysis and interpretation of data, Drafting or revising the article, Contributed unpublished essential data or reagents; PL, MZ, GH, HLi, Acquisition of data, Contributed unpublished essential data or reagents; SL, Conception and design, Acquisition of data, Contributed unpublished essential data or reagents; YZ, BC, Acquisition of data; MB, EKF, Analysis and interpretation of data; SXZ, Acquisition of data, Analysis and interpretation of data; HLu, Conception and design, Analysis and interpretation of data, Drafting or revising the article

## Author ORCIDs

Hua Lu, http://orcid.org/0000-0002-9285-7209

## Ethics

Animal experimentation: This study was performed in strict accordance with the recommendations in the Guide for the Care and Use of Laboratory Animals of the National Institutes of Health. All of the animals were handled according to approved institutional animal care and use committee (IACUC) protocols (#4257R) of Tulane University School of Medicine.

# Additional files

## Major datasets

The following dataset was generated:

| Author(s) | Year | Dataset title | Dataset URL | Database, license, and accessibility information |
|---|---|---|---|---|
| Zhou X, Lu H | 2016 | RNA sequencing analysis of NGFR ablation in both p53-positive and negative non-small-cell lung cancer cell lines | http://www.ncbi.nlm.nih.gov/geo/query/acc.cgi?acc=GSE79051 | Publicly available at the NCBI Gene Expression Omnibus (Accession no: GSE79051) |

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
