## [Decision Letter]

Thank you for submitting your article "Nerve Growth Factor Receptor Negates the Tumor Suppressor p53 as a Feedback Regulator" for consideration by *eLife*. Your article has been reviewed by two peer reviewers, and the evaluation has been overseen by a Reviewing Editor and Kevin Struhl as the Senior Editor.

The reviewers have discussed the reviews with one another and the Reviewing Editor has drafted this decision to help you prepare a revised submission.

Summary:

This paper reports that NGFR is a new transcriptional target of p53 that suppresses p53 by forming a ternary complex with p53 and MDM2 in the cell nucleus leading to augmented MDM2-mediated ubiquitination and degradation of p53. In addition, the paper demonstrates that NGFR inactivates p53 transcriptional activity independent of MDM2 through its ability to bind to the central DNA-binding domain of p53, thereby diminishing its ability to bind target promoters. That NGFR has a role in the nucleus and forms a negative feedback loop with p53 is a unique observation. Data are also presented showing that NGFR is frequently overexpressed in human glioblastomas; thus the NGFR-mediated negative regulation of p53 may have important implications in glioblastoma biology and therapeutics.

Essential revisions:

For this paper to be considered for *eLife* it is essential that the authors provide convincing demonstration that endogenously expressed NGFR is present in the nucleus and interacting with p53 there as outlined in Points 1 and 2.

1) Given that the membrane and nuclear NGFRs have completely different functions, it is of great importance to provide quantitative data on NGFR subcellular distribution and correlate them with the biological function. The fractionation results need to be validated with immunostaining data. The evidence of NGFR overexpression in human glioma was very clear but the images depicted in Figure 2 and Figure 2—figure supplement 1 could not clearly show NGFR subcellular distribution. It would be also of interest to know how NGFR distribution is regulated.

2) The use of H1299 cells expressing ectopic NGFR in Figure 5 is not appropriate since NGFR may be overexpressed and mis-localized as a result of the overexpression. H460 cells expressing ectopic NGFR in Figure 5 is not appropriate for the same reason. Demonstration of the endogenous interaction between NGFR, MDM2 and p53 in SK-N-SH cells shown in Figure 5 is central to the paper and this needs to be repeated using reciprocal IPs and cell fractionation. The fractionation experiments presented in Figure 5—figure supplement 1 using HCT116/p53+ and SK-MEL-147 cells show very little NGFR in the nucleus. Confocal microscopy is necessary to determine the proportion of endogenous NGFR present in the nucleus of these cells.

[Editors' note: further revisions were requested prior to acceptance, as described below.]

Thank you for submitting your article "Nerve Growth Factor Receptor Negates the Tumor Suppressor p53 as a Feedback Regulator" for consideration by *eLife*. Your article has been reviewed by two peer reviewers, and the evaluation has been overseen by a Reviewing Editor and Kevin Struhl as the Senior Editor. One of the two reviewers has agreed to reveal his identity: Zhi-Min Yuan (Reviewer #1).

The reviewers have discussed the reviews with one another and the Reviewing Editor has drafted this decision to help you prepare a revised submission. The manuscript is provisionally accepted, but the paper has been sent back for revision so that you can deal with the minor issues below as appropriate. The manuscript will be accepted upon its return without further review. The reason for this is that the paper will be published online quickly upon receipt, and it should be in final form at that time.

Summary:

The authors have carefully addressed most the comments raised by the reviewers by providing new data and additional information. The revised manuscript is significantly improved. The data collectively provide solid evidence indicating an intricate interaction among NGFR, p53 and MDM2, forming a negative feedback loop by which NGFR negatively regulates p53 function. The work is of significance as it uncovered a novel mechanism underlying the oncogenic function of NGFR. Considering the fact that NGFR is frequently overexpressed in a number of human cancers, the mechanistic information may carry therapeutic implication.

Essential revisions:

Reviewer #1:

The authors have carefully addressed most the comments raised by the reviewers by providing new data and additional information. The revised manuscript is significantly improved. The data collectively provide solid evidence indicating an intricate interaction among NGFR, p53 and MDM2, forming a negative feedback loop by which NGFR negatively regulates p53 function. The work is of significance as it uncovered a novel mechanism underlying the oncogenic function of NGFR. Considering the fact that NGFR is frequently overexpressed in a number of human cancers, the mechanistic information may carry therapeutic implication.

Figure 2 showed that knockdown of NGFR in carcinoma cells was associated with p53-dependent growth inhibition. Was p53 activated under the condition? If it was, what might be the potential mechanism?

Figure 7 to D: It is unclear why different target genes, PUMA VS p21, were used, and why none was shown in Figure 7.

Reviewer #2:

I am satisfied with the revisions made to the paper, with the additional data and with the authors' response to the reviewers' comments and criticisms. In my opinion, the revised paper is improved and merits publication.

---

## [Author Response]

*Essential revisions:*

*For this paper to be considered for eLife it is essential that the authors provide convincing demonstration that endogenously expressed NGFR is present in the nucleus and interacting with p53 there as outlined in Points 1 and 2.*

*1) Given that the membrane and nuclear NGFRs have completely different functions, it is of great importance to provide quantitative data on NGFR subcellular distribution and correlate them with the biological function. The fractionation results need to be validated with immunostaining data. The evidence of NGFR overexpression in human glioma was very clear but the images depicted in Figure 2 and Figure 2—figure supplement 1 could not clearly show NGFR subcellular distribution. It would be also of interest to know how NGFR distribution is regulated.*

Thanks for the suggestion. We carefully performed immunostaining and confocal microscopy analyses using a NGFR antibody (D4B3) from Cell signaling technology. This monoclonal antibody is produced by immunizing animals with a synthetic peptide corresponding to residues surrounding Arg198 which locates in the extracellular domain of human NGFR. As expected and in line with our fractionation results (Figure 5—figure supplement 1), we found that NGFR expression could be detected in the nucleus of all cells examined (Figure 5—figure supplement 1), although NGFR signals are more intensive in the cytoplasm and membrane than in the nucleus. While NGFR was originally found to be a transmembrane receptor in the neuronal cells, we have added new evidence that NGFR also resides in the nucleus and cytoplasm of cancer cells by cell fractionation and confocal microscopy analysis (Figure 5—figure supplement 1). These results, taken together with other results as shown in Figure 2–Figure 8), strongly support our conclusion that cancer cells hijack NGFR to overcome the cytotoxic effect of p53. Interestingly, the nuclear expression of NGFR increased when genotoxic insult was triggered by Doxorubicin (Figure 5—figure supplement 1). This elevated level of NGFR in the nucleus is likely due to p53 activation in response to Doxorubicin, as the NGFR gene is a p53 target gene. It is likely that the nuclear accumulated NGFR is to inactivate p53 thus inducing chemo-resistance of cancer cells, which is also consistent with our results shown in the Figure 7.

Given that p53 can be sequestered and degraded by MDM2 in the cytoplasm, it is reasonable that the cytoplasmic NGFR may also be involved in this process by binding to both p53 and MDM2 in the cytoplasm. Actually, we have some preliminary data from our ongoing project showing that NGFR retains p53 in the cytoplasm by associating with another novel p53 regulatory protein, which will be described in a separate report in the near future.

It is intriguing and worthwhile to figure out the mechanism underlying the intracellular translocation of NGFR in cancer cells. By treating melanoma cells with DNA damaging agents, we observed that the nuclear portion of NGFR increased in some of the cells by confocal microscopy analysis. However, to elucidate the exact molecular basis, more precise and quantitative experiments need to be designed and a multitude of cancer-associated signals will be explored. I hope that you would allow us to address this mechanistic question on how NGFR is transported to the nucleus in an independent study in the future

By sharing these incomplete studies with you and the reviewers, I want to let you know that the relationship between subcellular localization of NGFR and its tumorigenic role might be more complicated than one could imagine in primary human cancers. Also, it is more challenging to use currently available anti-NFFR antibodies for detailed IHC analysis of primary human cancers in order to distinguish NGFR in different subcellular compartments and to correlate its subcellular localization with different stages of human tumors or different types of tumors. Nevertheless, our studies as shown in this manuscript do confirm that nuclear NGFR can bind to p53 and MDM2, consequently inactivating p53 by directly inhibiting its transcriptional activity and facilitating MDM2-dependent ubiquitination and degradation.

*2) The use of H1299 cells expressing ectopic NGFR in Figure 5 is not appropriate since NGFR may be overexpressed and mis-localized as a result of the overexpression. H460 cells expressing ectopic NGFR in Figure 5 is not appropriate for the same reason. Demonstration of the endogenous interaction between NGFR, MDM2 and p53 in SK-N-SH cells shown in Figure 5 is central to the paper and this needs to be repeated using reciprocal IPs and cell fractionation. The fractionation experiments presented in Figure 5—figure supplement 1 using HCT116/p53+ and SK-MEL-147 cells show very little NGFR in the nucleus. Confocal microscopy is necessary to determine the proportion of endogenous NGFR present in the nucleus of these cells.*

In response to these suggestions, we have performed more experiments. To confirm our data presented in the Figure 5 (SK-N-SH cell line) and 4I (NGFR-stably expressed H460 cell line), we conducted cell fractionation and reciprocal IP assays using another cell line, SK-MEL-147 melanoma cell line, that also highly expresses NGFR. As shown in Figure 5, the nuclear extracts were subjected to IP using anti-NGFR, anti-p53 or anti-MDM2. In line with the former results (Figure 5), NGFR could be co-immunoprecipited with p53 and MDM2 in a reciprocal pattern.

As mentioned in our response to the point 1, we carefully designed confocal microscopy analysis by using an antibody (D4B3, Cell signaling tech) recognizing the N-terminal extracellular domain of NGFR. As shown in this experiment, NGFR is detected in all of the cellular compartments including the nucleus, cytoplasm, and membrance of all the SK-MEL-147 melanoma cells examined, although less NGFR was found in the nucleus than in the cytoplasm and membrane (Figure 5—figure supplement 1).

[Editors' note: further revisions were requested prior to acceptance, as described below.]

*Reviewer #1:*

*The authors have carefully addressed most the comments raised by the reviewers by providing new data and additional information. The revised manuscript is significantly improved. The data collectively provide solid evidence indicating an intricate interaction among NGFR, p53 and MDM2, forming a negative feedback loop by which NGFR negatively regulates p53 function. The work is of significance as it uncovered a novel mechanism underlying the oncogenic function of NGFR. Considering the fact that NGFR is frequently overexpressed in a number of human cancers, the mechanistic information may carry therapeutic implication.*

*Figure 2 showed that knockdown of NGFR in carcinoma cells was associated with p53-dependent growth inhibition. Was p53 activated under the condition? If it was, what might be the potential mechanism?*

Yes, p53 was activated upon knockdown of NGFR as shown Figure 3. The mechanisms are shown in Figure 4–Figure 6, increasing p53 stability and releasing p53 from the NGFR inhibition of its promoter-binding.

Figure 7: It is unclear why different target genes, PUMA VS p21, were used, and why none was shown in Figure 7.

We can explain it. First, the bands of p21 and PUMA on an SDS gel are often close to each other, sometimes when the concentration of the polyacrylamide gel is not optimal to separate small proteins. In this case, we only detected either p21 or PUMA expression as an indication of p53 activity, since knockdown of NGFR has been shown to activate the p53 pathway markedly in Figure 3 with not only the WB data, but also the RNA-seq data. Because of the convincing reproducibility of these findings each time in each of our assays, we thus thought it is unnecessary to show p21 or Puma in all of the blots, such as Figure 7. However, convincingly, knockdown of NGFR can sensitize these cancer cells to chemotherapeutic agents, such as cisplatin or doxorubicin, as tested here. This sensitization is more apparent in p53-positive cancer cells, though Doxorubicin could also induce p21 independently of p53.